# External Validation with Accuracy Confounders of VCO_2_-Derived Predicted Energy Expenditure Compared to Resting Energy Expenditure Measured by Indirect Calorimetry in Mechanically Ventilated Children

**DOI:** 10.3390/nu14194211

**Published:** 2022-10-10

**Authors:** Panagiotis Briassoulis, Stavroula Ilia, Efrossini Briassouli, George Briassoulis

**Affiliations:** 1Pediatric Intensive Care Unit, University Hospital, School of Medicine, University of Crete, 71110 Heraklion, Greece; 2Department of Anaesthesiology, Attikon University Hospital, School of Medicine, National and Kapodistrian University of Athens, 12462 Athens, Greece; 3Postgraduate Program “Emergency and Intensive Care in Children Adolescents and Young Adults”, School of Medicine, University of Crete, Voutes, 71003 Heraklion, Greece; 4Infectious Diseases Department “MAKKA”, First Department of Paediatrics, “Aghia Sophia” Children’s Hospital, Athens Medical School, 11527 Athens, Greece

**Keywords:** children, indirect calorimetry, resting energy expenditure, accuracy, critical care, prediction equations

## Abstract

Optimal energy provision, guided by measured resting energy expenditure (REE) and determined by indirect calorimetry (IC), is fundamental in Intensive Care Units (ICU). Because IC availability is limited, methods to predict REE based on carbon dioxide production (VCO_2_) measurements (REE_VCO2_) alone have been proposed as a surrogate for REE measured by IC (REE_IC_). The study aimed at externally and internally validating the accuracy of the REE_VCO2_ as an alternative to REE_IC_ in mechanically ventilated children. A ventilator’s integrated gas exchange module (E-COVX) was used to prospectively measure REE_IC_ and predict REE_VCO2_ on 107 mechanically ventilated children during the first 24 h of admission. The accuracy of the REE_VCO2_ compared to REE_IC_ was assessed through the calculation of bias and precision, paired median differences, linear regression, and ROC analysis. Accuracy within ±10% of the REE_IC_ was deemed acceptable for the REE_VCO2_ equation. The calculated REE_VCO2_ based on respiratory quotient (RQ) 0.89 resulted in a mean bias of −72.7 kcal/day (95% limits of agreement −321.7 to 176.3 kcal/day) and a high coefficient of variation (174.7%), while 51.4% of the calculations fell outside the ±10% accuracy rate. REE_VCO2_ derived from RQ 0.80 or 0.85 did not improve accuracy. Only measured RQ (Beta 0.73, *p* < 0.001) and no-recorded neuromuscular blocking agents (Beta −0.13, *p* = 0.044) were independently associated with the REE_VCO2_−REE_IC_ difference. Among the recorded anthropometric, metabolic, nutrition, or clinical variables, only measured RQ was a strong predictor of REE_VCO2_ inaccuracy (*p* < 0.001). Cutoffs of RQ = 0.80 predicted 89% of underestimated REE_IC_ (sensitivity 0.99; specificity 0.89) and RQ = 0.82 predicted 56% of overestimated REE_IC_ (sensitivity of 0.99; specificity 0.56). REE_VCO2_ cannot be recommended as an alternative to REE_IC_ in mechanically ventilated children, regardless of the metabolic, anthropometric, or clinical status at the time of the evaluation.

## 1. Introduction

Indirect calorimetry (IC) is recommended by the American Society for Parenteral and Enteral Nutrition (ASPEN) and the European Society of Paediatric and Neonatal Intensive Care (ESPNIC) to measure resting energy expenditure (REE) and guide optimal energy provision in mechanically ventilated adult [1] and pediatric [2] patients. Indirect calorimetry (IC) is based on the volumetric measurement of oxygen consumption (VO_2_) and carbon dioxide production (VCO_2_), deriving REE by Weir’s equation [3]. Recent technical developments allow integrated ventilators’ IC modules to measure breath-by-breath resting energy expenditure (REE_IC_) in mechanically ventilated patients [4,5].

Despite recent recommendations, REE_IC_ is not measured routinely in most mechanically ventilated patients because of the lack of equipment, cost, and expertise to conduct IC and analyze results, while equations used to predict REE are inaccurate [6,7]. Suboptimal feeding, however, can lead to malnutrition, a longer duration of mechanical ventilation, and increased morbidity and mortality [8].

Novel equations to estimate REE based on VCO_2_ (REE_VCO2_) derived from measurements of exhaled gas volume and CO_2_ concentrations have been recently proposed as a surrogate for IC in pediatric and adult patients [9,10]. The REE_VCO2_ predictive equation assumes a fixed respiratory quotient (RQ) value of (0.85) [11], (0.89) [10], or equal to an estimated RQ from oxidation of energy substrates [12]. However, the variability of RQ might interfere with the accuracy of the REE_VCO2_ calculation [9,11].

The purpose of this study is to externally validate the accuracy of the REE_VCO2_ and whether it could be considered as an alternative to REE_IC_, using a ventilator’s integrated IC module for both methods. A secondary objective is to identify and internally validate clinical or metabolic factors that might influence the performance of the REE_VCO2_ equation in mechanically ventilated children.

## 2. Materials and Methods

### 2.1. Study Design

Mechanically ventilated critically ill children consecutively admitted to the academic Pediatric Intensive Care Unit (PICU) at the University Hospital, School of Medicine, University of Crete, Heraklion, from June 2015 through June 2018 were enrolled in the study. The Ethics Committee of the Institutional Review Board approved the study (approval ID14494/2011/9-1-2012). All of the data were de-identified, and the parents or guardians provided informed written consent. The present study was conducted in accordance with the principles of the Declaration of Helsinki (last revised guidelines from 2013), following the International Conference on Harmonization (ICH)/Good Clinical Practice (GCP) standards [13].

Inclusion criteria: Hemodynamically stable, adequately sedated (Ramsey > 3), mechanically ventilated patients with a Fractional Inspired Oxygen (FiO_2_) < 60%, a respiratory rate below 35 breaths-per-minute, and an endotracheal tube (ET) leak below 10% [inspiratory tidal volume (TVi) − expiratory tidal volume (TVe)/inspiratory TV × 100] were eligible for the study [14]. Exclusion criteria: (1) Patients expected to be extubated within 24 h of admission; (2) inborn errors of metabolism or primary endocrine disorders; (3) unexpected interruption of the measurement (destabilization, need for intervention in the ventilation settings, or other).

### 2.2. Clinical Data

At the time of each metabolic measurement, admission diagnosis, ventilatory settings, blood pressure, heart rate, sedation level by Ramsey scale, and main sedatives and vasoactive agents or inotropes were recorded. The last recorded temperature on a patient’s vital signs flowchart just before the REE measurement was documented. The severity of illness was assessed using the PRISM-III and the PELOD-2 scores [15], and the amount of care was assessed using the Therapeutic Intervention Scoring System (TISS) [16]. The energy intake was calculated from recorded intake of enteral or parenteral nutrition and glucose-containing maintenance fluids. Underfeeding and overfeeding were defined as energy intake of <90% and >110% of measured REE, respectively.

### 2.3. Anthropometry 

The following anthropometric parameters were identified: age, sex, actual weight, ideal weight, height, and body mass index (BMI). The weight was measured using calibrated electronic bed scales. The ideal weight was defined as the weight for the 50th percentile of the actual height of each patient. The BMI was calculated as kg/m^2^. The standard deviation scores, known as z-scores, of weight, height, and BMI for sex and age were calculated using WHO and CDC calculators [17]. Malnutrition indices were derived from the BMI for age and sex z-scores obtained at admission. Underweight was defined as a BMI z-score < −1.644, normal weight as −1.644 ≤ BMI z-score < 1.036, overweight as 1.036 ≤ BMI z-score < 1.644, and obesity as BMI z-score ≥ 1.644. 

### 2.4. Indirect Calorimetry

An integrated gas exchange module (E-COVX) into the ventilator (Carescape R860, GE Healthcare, Milwaukee, WI, USA) was used to measure REE through indirect calorimetry during the ICU’s first 24 h. This module is able to reliably record spirometry and metabolic indices as early as 5 min after suctioning at different modes of ventilation [4,5]. It has no mixing chamber, and the sampling takes place with every breath. It has a fast differential paramagnetic O_2_ and infrared CO_2_ analyzer and a pneumotachograph housed in a connector, which measures inspired and expired volumes. In the P-Lite (15–300 mL) or D-Lite (>300 mL) flow sensor, located proximate to the Y-piece to the patient’s ET tube, the flow measurement is based on the pressure drop across a special proprietary turbulent flow restrictor. It uses mathematical integration of flow and time-synchronized continuous gas sampling to provide data. The gas sample is continuously drawn from the connector to the gas analyzer unit of the module. Both O_2_ and CO_2_ measures are based on the side-stream principle. E-COVX relies on tidal volume measurement for VO_2_ calculation. The pneumotachograph derives the tidal volume from the pressure difference across a fixed orifice, potentially influenced, therefore, by acute changes of resistance in the spirometry tubing and undetected leaks in the system. We consistently used a heat- and moisture-exchange filter alone, avoiding heated water bath humidification, followed by regular checks on the spirometry tubing and checks for tidal volume consistency between the module and the ventilator. 

The measurements were taken between 9 am and 12 pm when there had been a minimum of 45 min with no major physical activity, such as physiotherapy or dressing change. After an initial 20 min stabilization period, REE was measured for 30 min, during which time there was no interference with the child. The module uses the modified Weir formula (Equation (1)) and displays a 5 min average for REE but can display the 1 min averages with the S/5 Collect 1.0 Software (Datex-Ohmeda, GE Healthcare, Waukesha, WI, USA).
REE_IC_ (kcal/day) = [3.941 × VO_2_ + 1.106 × VCO_2_] × 1440(1)

The steady state was defined as a period of at least 5 min with less than 10% fluctuation in VO_2_ and VCO_2_ and less than 5% fluctuation in RQ, which is the VCO_2_/VO_2_ ratio. Measurements with RQ outside the physiologic range (>1.3 or <0.67) were excluded. 

### 2.5. VCO_2_-Derived REE

For the VCO_2_-derived formula, REE_VCO2_ was calculated using VCO_2_ values measured by indirect calorimetry by assuming an RQ  =  0.89, as has been previously proposed for mechanically ventilated children [10] (Equation (2)). For comparison, we analyzed Equation (2) with an expanded value of 5.534 instead of 5.5 [18].
REE_VCO2_ (kcal/day) = 5.5 × VCO_2_ (L/min) × 1440(2)

The RQ values of 0.85, 0.80, and 0.89, using original equations derived from the modified Weir formula, which have been used in published validation research [19], were also examined (Equation (3)). As a control, we calculated Equation (3) using the IC-derived individual RQ (RQ_IC_) along with the individual VCO_2_ values, resulting in a VO_2_ alone REE_VCO2_–REE_IC_ difference (Equation (3)).
REE_VCO2_ (kcal/day) = ((5.5 × (VCO_2_/0.89 or 0.85 or 0.80 or RQ_IC_)) + (1.76 × VCO_2_) − 26)(3)

Basal metabolism was calculated based on the Schofield equation [20]. Hypometabolism and hypermetabolism were defined as measured REE of <90% and >110% of basal metabolic rate as predicted by Schofield’s equation, respectively [20]. The patients were stratified for subsequent subset analysis according to their nutritional status as per BMI for age and sex z-scores.

### 2.6. Statistical Analysis

The normality of the distribution was examined using the Shapiro–Wilk test. The descriptive data are reported as means and standard deviation (SD) or median and interquartile range (IQR) in case of skewed distributions or as frequencies and percentages when appropriate. The agreement of REE_VCO2_ with REE_IC_ was assessed through the calculation of bias and precision. Bias was defined as the mean difference between the measurements obtained from REE_VCO2_ and REE_IC_. Precision was defined by the 95% limits of the agreement, including both systematic (bias) and random error. The paired median differences were also calculated, and relative variability (dispersion) and repeatability were assessed by calculating the coefficient of variation (CV), which is the ratio of the standard deviation to the mean of the population. Clinically significant percentage error (REE_VCO2_–REE_IC_)/REE_IC_ (%) was considered a difference of ≥±10%. The reliability was assessed by the intraclass correlation coefficient (ICC), calculated using the two-way mixed (Cronbach’s alpha). ICC was interpreted as follows: below 0.50: poor; between 0.50 and 0.75: moderate; between 0.75 and 0.90: good; above 0.90: excellent [21]. A linear regression model (backward method) was adopted to examine whether any of the recorded anthropometric, clinical, and metabolic variables are independently associated with the REE_VCO2_–REE_IC_ difference. We first used univariate models to test if any of the studied variables were related to the REE_VCO2_–REE_IC_ difference, with just one explanatory variable at a time; afterward, all variables that had shown a relaxed *p*-value of less than or equal to 0.1 were included in the multivariate models. To evaluate factors affecting REE_VCO2_ accuracy, the areas under the receiver operating characteristic curves (AUROCs) for variables significantly predicting REE_VCO2_–REE_IC_ differences outside the 10% clinically acceptable estimations were calculated. A two-sided significance level of 0.05 was used for statistical inference. Statistical analysis software (version 28, SPSS, Chicago, IL, USA) was used for all analyses, and GraphPad Prism 9.0 (GraphPad Software, Inc., San Diego, CA, USA) was used for the Bland–Altman analyses and illustrations.

## 3. Results

### 3.1. Study Population

During the study period, 486 patients were admitted to the PICU, of which 121 were eligible for inclusion (Figure 1). However, 14 patients were not enrolled due to logistic reasons (*n* = 10), technical reasons (*n* = 3), or no informed consent (*n* = 1). The demographic, anthropometric, and clinical characteristics of the enrolled 107 patients are shown in Table 1. Twenty-three patients were underweight (21.5%), 10 were overweight (9.3%), and 27 obese (25.2%). All of the patients received sedation and/or analgesia. The energy received on PICU day 1 was provided by continuous enteral nutrition and non-nutritional energy sources (*n* = 102, 95.32%) or parenteral nutrition (*n* = 5, 4.67%) by clinicians who were blinded to the methodology of this study.

### 3.2. Performance of the REE_VCO2_ Equation

Paired REE_VCO2_ (910 (666; 1389) kcal/day)–REE_IC_ (999 (703; 1416) kcal/day) median differences were significant (−71.01 (−92.9; −49.9), *p* < 0.001). Although the REE_VCO2_ reliability, as assessed by the ICC, was excellent (Cronbach’s alpha, 0.979, *p* < 0.001), 51.4% of the calculations fell outside the 10% accuracy rate, especially among underweight (61%) or obese patients (55.6%) The inaccuracy profile varied from underestimation (45.8%) to overestimation (5.6%) (Table 2).

The calculated REE_VCO2_ resulted in a mean bias of −72.7 with a wide dispersion of values as expressed by the 95% limits of agreement (precision) (−321.7 to 176.3 kcal/day) and a high coefficient of variation (174.7%). The mean percentage bias (−6.57 ± 10.4%), 95% limits of agreement (−26.9 to 13.8%), and 95% confidence intervals (−8.6 to −4.6%) are presented in the Bland–Altman plot of Figure 2.

### 3.3. REE_VCO2_ Using Different RQ Values

The reliability of REE_VCO2_ using different RQ values remained excellent (RQ = 0.89, Cronbach’s alpha 0.980 for RQ = 0.89, 0.981 for RQ = 0.85, and 0.982 for RQ = 0.80, all *p* < 0.001). The median (IQR) of REE_VCO2_ using arbitrary RQ values is presented in Table 3. REE_VCO2_ derived from an RQ of 0.80 had the lowest median difference and bias (−8.7 kcal/day), an accuracy of 57% (within 10% of REE_IC_ values), and the highest dispersion of values (CV 1427%). Equations using an RQ of 0.89 had the highest bias and significant median differences (*p* < 0.001), while accuracy was lower than 50% for all equations using an RQ of 0.85 or 0.89. The control equation using the individual RQ_IC_ values presented the best accuracy (89.7%), but a significant median difference (*p* < 0.001), CV of 266%, and a bias of −38.48 (limits of agreement −239.1; 162.2) kcal/day (Appendix A, Appendix A).

### 3.4. Factors Affecting the REE_VCO2_ Accuracy

There were no significant differences in the ±10% accuracy rates of REE_VCO2_ among patients with different nutrition (under- or overfeeding, *p* = 0.57) or metabolic patterns (hypo- or hypermetabolism, *p* = 0.18). Bivariate analysis showed that REE_VCO2_–REE_IC_ difference (r^s^ = 0.35, *p* = 0.013) and RQ (r^s^ = 0.32, *p* = 0.022) correlated with lactate but not with BMI z-scores, age, metabolic status, TISS, PRISM, PELOD, heart or respiratory rate, blood pressure, temperature, blood gases, glucose, albumin, C-reacting protein, or energy intake.

In a linear regression model (stepwise, backward method), only measured RQ (Beta 0.73, *p* < 0.001) and no-recorded neuromuscular blocking agents (Beta −0.13, *p* = 0.044) were independently associated with the REE_VCO2_–REE_IC_ difference. The non-linear least squares regression fit of REE_VCO2_ predictions expressed as paired REE_VCO2_–REE_IC_ differences over the range of recorded RQ values (polynomial (quadratic) equation) is shown in the scatterplot in Figure 3. The number of sedatives, opioids, or vasoactive drugs did not affect the difference between the two methods. Additionally, none of the patients’ demographics, BMI nutrition status (overweight, obesity), diagnostic category, the severity of illness, temperature, heart rate, blood gases, or CRP was independently associated with the REE_VCO2_–REE_IC_ difference.

In a ROC analysis, only the measured RQ was a strong predictor of REE_VCO2_ ^RQ 0.89^ inaccuracy, underestimating REE_IC_ for more than −10% (AUROC 0.991 (95%CI 0.975–1.0), *p* < 0.001) (Table 4).

The “optimal” cutoff point of RQ for the best sensitivity-specificity combination was calculated by the Youden index (J) and confirmed by the Closest to (0, 1) Criteria (ER) [22]. Cutoffs of RQ = 0.80 reached a sensitivity of 0.99 and specificity of 0.89 for 89% of REE_VCO2_ values underestimating REE_IC_ measurements (Youden index 0.89, predicting 89% of underestimated measurements). The accuracy rates were not affected by other recorded metabolic, anthropometric, or clinical variables at the time of measurements (Figure 4).

Regarding the smaller proportion of +10% inaccuracy, measured RQ (AUROC 0.804 (95%CI 0.643–0.966), *p* = 0.013) and a high PRISM III score (AUROC 0.819 (95%CI 0.646–0.992), *p* = 0.009) were strong predictors of REE_VCO2_ overestimating REE_IC_ for more than +10% (Table 5).

Cutoffs of RQ = 0.82 (sensitivity 0.99, specificity of 0.56, Youden index 0.57) and of PRISM II = 16.5 (sensitivity 0.83, specificity 0.85, Youden index 0.68) predicted 56% and 68% of REE_VCO2_ values overestimating REE_IC_ measurements, respectively (Figure 5).

## 4. Discussion

In this study, we aimed to externally validate the accuracy of the REE_VCO2_ and internally identify factors that might influence the performance of the REE_VCO2_ in mechanically ventilated children. First, we showed that more than half of the REE_VCO2_ calculations fell outside the 10% accuracy rate compared to the measured REE_IC_. The calculated REE_VCO2_ resulted in a wide dispersion of values as expressed by the extended 95% limits of agreement and a high coefficient of variation. Second, we showed that only RQ and avoidance of neuromuscular blocking agents were independently associated with the REE_VCO2_–REE_IC_ difference. Second, we showed that RQ <0.80 or >0.82 significantly underestimates or overestimates REE_IC_ measurements and that accuracy rates are not affected by other recorded metabolic, anthropometric, or clinical variables. Finally, we demonstrated that the accuracy of REE_VCO2_ did not improve using RQ values of 0.80, 0.85, or 0.89. The wide dispersion of the values remaining when we substituted the fixed by the individual RQ recordings disclosed the key role of online measuring VO_2_ in calculating REE in mechanically ventilated patients. These results indicate that the REE_VCO2_ predictive equation cannot be recommended as an alternative to measured REE_IC_ in mechanically ventilated children, regardless of the metabolic, anthropometric, or clinical status at the time of the evaluation.

In a cohort of adult ICU patients, the limits of agreement in the Bland–Altman plots (−356 to +314 for REE_VCO2_0.85_ and −367 to +272 for REE_VCO2_FQ__ derived from Food Quotient) were close to the levels found in our study (−321.7 to +176.3 kcal/day) [12]. This wide range of 95% limits of agreement does not support the findings of an older validation dataset reporting narrow limits of agreement for the REE_VCO2_ equation [10]. Using a mass spectrometer and the REE_VCO2_ = 5.534 × VCO_2_ (L/min) × 1440 equation in a cardiac, pediatric cohort, Mouzaki et al. showed that the REE_VCO2_–REE_IC_ agreement was biased during a 72 h period following cardiopulmonary bypass (mean percentage error 11% ± 7%) [18]. Other studies have also demonstrated a low level of agreement of REE_VCO2_ vs. REE_IC_ readings (limits of agreement >1000 kcal/day) [19], recommending indirect calorimetry as the method for REE assessment in critically ill patients [18,19]. In our study, the low ±10% accuracy rates of the recommended [7] (48.6%) or other commonly used [19,23] REE_VCO2_ equations (43–57%) were worse than those reported in adult studies (61–78%) using different equipment, ventilators, and/or methodology [9,12]. A high accuracy rate of 89% obtained by Rousing using VCO_2_ derived from the metabolic monitor for both equations was characterized by large variations in minute ventilation, increasing the discrepancy between metabolic production and pulmonary uptake or CO_2_ excretion [11]. Using individual RQ_IC_ and VCO_2IC_ values obtained through IC for both equations, we also recorded a high ±10% accuracy rate (89.7%). However, this is the first time to show the importance of the VO_2IC_ variability in estimating REE of individuals in ICU. Because the time constant for VO_2_ equilibration is much shorter (2–3 min) than the VCO_2_ (10–20 min), variations in ventilation will affect the REE_IC_ differently, incorporating the VO_2IC_ in the equation, compared to the REE_VCO2_, missing the breath-by-breath metabolic marker VO_2IC_ quick measurements [24]. The wide scattering of VO_2IC_ values in our control group verifies the vulnerability of the REE_VCO2_ in estimating the patient’s metabolism changes, precluding its use as an alternative to REE_IC_ in mechanically ventilated patients. 

Mehta et al. developed a simplified Weir equation, which performed well when validated using a separate group of critically ill children admitted to a second ICU [10]. They used an RQ of 0.89 defined by macronutrient administration, describing a mean percentage bias (limits for agreement) of −0.65% (−14.4 to 13.1%) kcal/day. The RQ of 0.89 was based on a cohort of younger patients with a median age of 2.34 for the derivation and 0.46 years for the validation datasets [10]. Using the same mean RQ of 0.89 in a cohort of neonates and infants of 5.1 months [18], Mouzaki et al. demonstrated a wide dispersion of values, resulting in approximately 50% of all observations falling outside the RQ limits of 0.8 and 1.0 and wide limits of agreement [18]. Using the same proposed equation based on RQ = 0.89, but in an older cohort of patients (mean age 9.2 years ), we demonstrated a significantly higher mean percentage bias −6.57 ± 10.4% (−26.9 to 13.8%) kcal/day, with a measured median RQ of 0.81 (IQR 0.75 to 0.91), similar to an RQ of 0.81, previously reported in adult patients [11]. Importantly, we demonstrated that the accuracy of REE_VCO2_ did not improve using RQ values of 0.80, 0.85, or 0.89. Although the RQ in our study presented a trend for lower recordings in older patients, it also showed a wide spread of values across all age groups. This wide dispersion of values remained, even when we substituted the fixed by the individual RQ recordings, disclosing the key role of measuring VO_2_ in calculating REE in mechanically ventilated patients. However, the value of RQ, which is the main predictor of the REE_VCO2_ accuracy, will not be available in clinical practice, leading more than half of the patients to under or overfeeding.

This is the first study prospectively incorporating that measured by the integrated to ventilator gas exchange module (E-COVX^®^) VCO_2_ readings in the REE_VCO2_ predictive equation, leaving that measured by indirect calorimetry RQ values as the only determinant of the REE_VCO2_–REE_IC_ difference. In a retrospective adult study, the REE_VCO2_ equation was analyzed using the VCO_2_ measured by the Deltatrac^®^ Metabolic Monitor, leaving the assumed RQ = 0.85 as the only determinant of the REE_VCO2_ accuracy [12]. Our results agree with those of a previous study, showing that the main determinant of bias of the REE_VCO2_ calculations is the RQ, especially among those with an RQ < 0.80 [18]. Except for lactate, we did not find any correlation between RQ and the clinical, metabolic, or nutrition indices. In our study, energy intake was not associated with REE_VCO2_ inaccuracy. In accordance with our methodology, Oshima et al. compared REE_VCO2_ with REE_IC_ with the VO_2_, VCO_2_, and REE measurements obtained exclusively from IC [12]. In contrast to the pediatric REE_VCO2_ equations using an RQ of 0.89 [10], they used an RQ of 0.85 or an RQ related to nutritional intake [9]. They confirmed that RQ is neither a reliable indicator of the feeding status nor strongly associated with non-nutritional factors and concluded that REE_VCO2_ could not be considered an alternative to REE_IC_. It has also been shown that feeding status is a factor that did not need to be considered in the assessment of RQ variability since no appreciable thermogenic effect of feeding occurs in continuously fed critically ill patients [25], and the RQ is not related to the feeding status or to ventilatory and acid-base disturbance [26]. Importantly, the VCO_2_ and VO_2_ slopes change between non-survivors and survivors [27], with the VO_2_/lactate ratio being significantly higher in survivors [28]. In accordance with our previous research, this study showed that RQ is highly variable and unpredictable in mechanically ventilated children [29], limiting its validity as an indicator of energy substrate oxidation [22], misleading energy provision targets [30], and enhancing the risk of underfeeding and overfeeding [31]. Using ROC analysis, we detected a bidirectional interference of RQ in predicting REE_VCO2_ inaccuracy in mechanically ventilated children since 89% of patients with RQ < 0.80 underestimated REE_IC_ by more than 10%, and 57% of those with RQ > 0.82 overestimated REE_IC_ by more than 10%. 

Although this study’s sample size is small, it is comparable to other single-center studies. Additionally, this is a prospective cross-sectional study, while the timing of the indirect calorimetry measurements reflects the acute only metabolic phase of illness. Although insignificant differences have been previously demonstrated during the critical first week of critical illness [12], other studies showed a pattern of early longitudinal repression of bioenergetics, the persistence of which is associated with poor outcomes [7,22,32]. Although the contribution of the acute-phase endogenously produced energy cannot be measured, the wide range of metabolic alterations, which cannot be accurately predicted, demand the daily use of indirect calorimetry to guide personalized nutrition in mechanically ventilated children to prevent cumulative energy balance excesses and deficits [33]. Along with the non-measured endogenously produced energy and the changing feeding pattern of the first ICU Day, the confirmed unreliability of the RQ as an indicator of the feeding status did not allow us to obtain a reliable RQ derived from Food Quotient.

Another limitation of the study is that the measurements of VCO_2_ in the present study were from breath-by-breath indirect calorimetry through an integrated ventilator’s gas exchange module and not a product of a ventilator expiratory CO_2_ concentration and volume, as in previous studies [9]. The within-device reliability of the VCO_2_ measurements of the E-COVX^®^ has been shown to be 1.5 mL/min at FiO_2_ 21–50% [34] with an accuracy rate from 7.2 to −5.2% [14], compared to a ±9% accuracy of the calculated expiratory VCO_2_ [12]. Accordingly, it seems that the VCO_2_ product calculated by a different ventilator might not improve the accuracy of the indirect calorimetry derived VCO_2_ and thus influence the results of this study.

## 5. Conclusions

The wide limits of agreement and high percentage error suggest that the REE_VCO2_ equation has limited clinical utility and that indirect calorimetry remains the gold standard to guide nutrition therapy in mechanically ventilated children. Since the accuracy of REE_VCO2_ may not be improved using various arbitrary or individual RQ values, online VO_2_ measurements are critical in accurately calculating REE in mechanically ventilated patients. The predicted equation based on VCO_2_ measurements alone cannot be recommended as an alternative to REE_IC_, regardless of the metabolic, anthropometric, or clinical status at the time of the evaluation. A new generation of user-friendly calorimeters, cost-effective, incorporated into ventilators’ hardware and software, are a one-way street to overcome the current limitations in reliably measuring real-time REE in an intensive care setting. 

## Figures and Tables

**Figure 1 nutrients-14-04211-f001:**
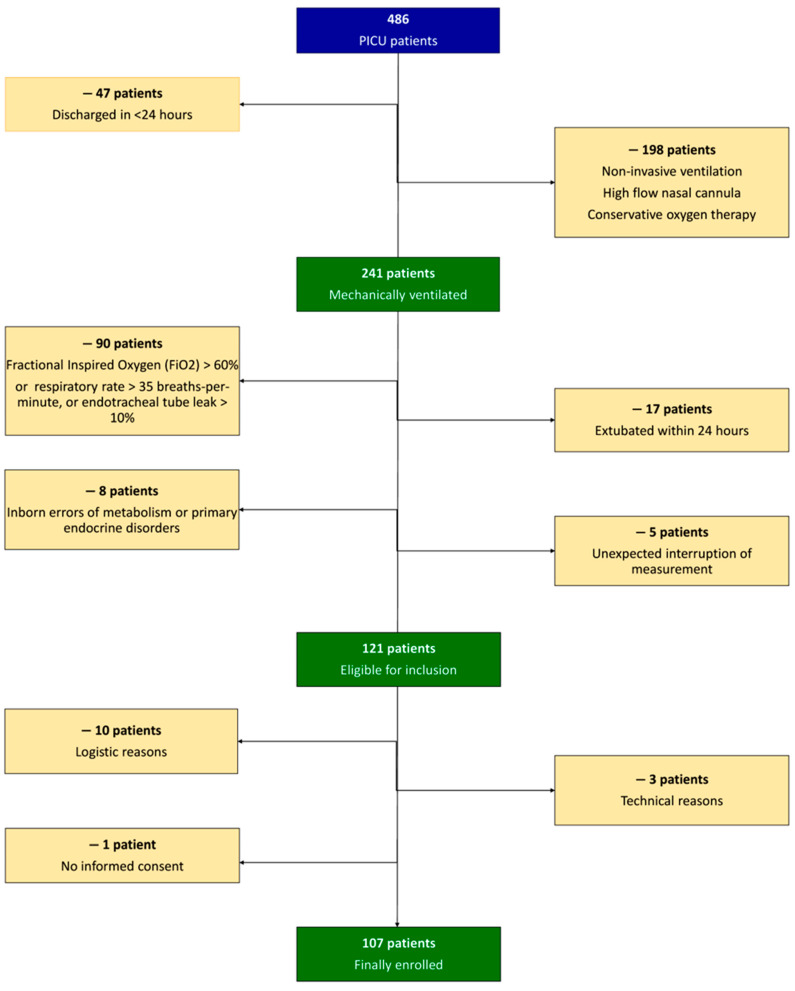
Flow chart.

**Figure 2 nutrients-14-04211-f002:**
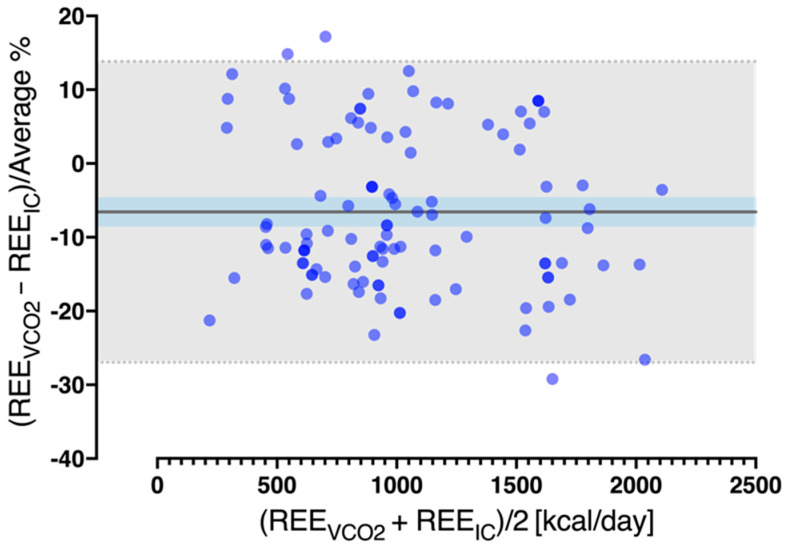
Bland–Altman plot whereby resting energy expenditure (REE) based on volumetric carbon dioxide production (VCO_2_) measurements (REE_VCO2_) alone is compared to REE measured by IC (REE_IC_) at ICU Day-1. The solid line indicates the percentage of agreement bias (%), and the light shade with the fine dotted lines indicates the limits of agreement (bias ± (1.96 × SD) = precision). Dark shade represents the 95% confidence intervals of the mean (bias).

**Figure 3 nutrients-14-04211-f003:**
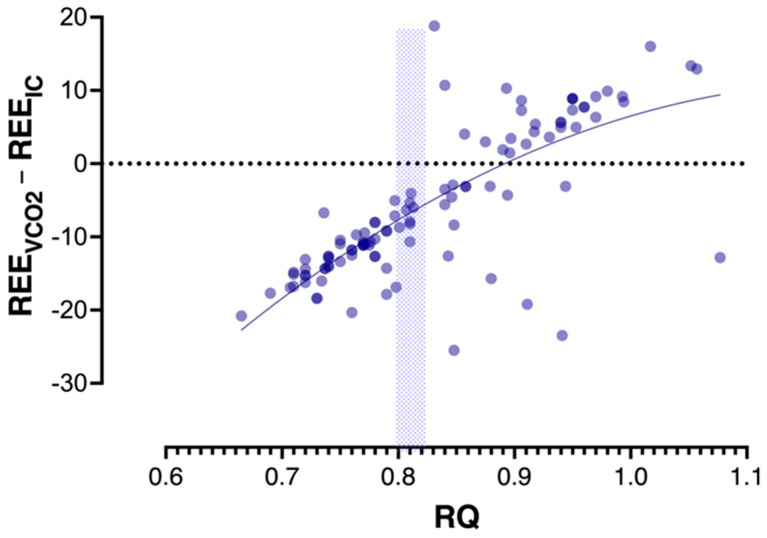
Scatterplot of non-linear least squares regression fit of resting energy expenditure (REE) based on volumetric carbon dioxide production (VCO_2_) measurements (REE_VCO2_) predictions expressed as paired REE_VCO2_–REE calculated by indirect calorimetry (IC) (REE_IC_) differences over the range of recorded respiratory quotient (RQ) values (polynomial (quadratic) equation). Analysis of areas under the receiver operating characteristic curves (AUROCs) revealed that cutoffs of RQ = 0.80 predict 89% of REE_VCO2_ underestimating REE_IC_ (sensitivity 0.99; specificity 0.89) and RQ = 0.82 predict 56% of REE_VCO2_ overestimating REE_IC_ (sensitivity of 0.99; specificity 0.56) (areas outside the shaded rectangle).

**Figure 4 nutrients-14-04211-f004:**
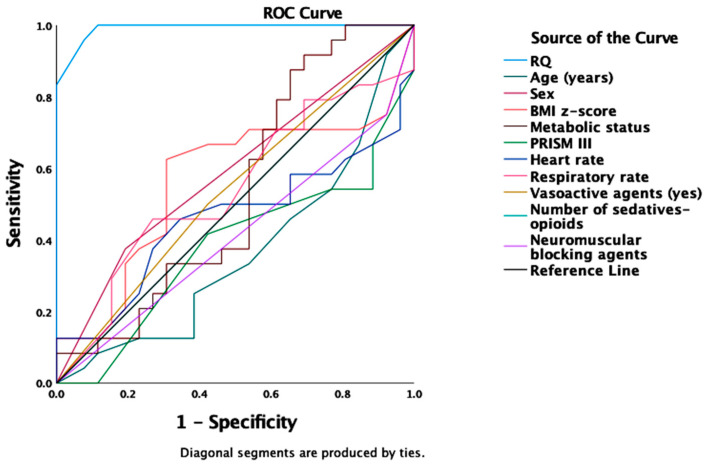
An Area Under the Receiver Operating Characteristic Curve (AUROC) for predicting REE_VCO2_ ^RQ 0.89^ underestimating REE_IC_ for more than −10%. Among various metabolic, demographic, and clinical variables, only the measured RQ was a strong predictor of REE_VCO2_ ^RQ 0.89^ inaccuracy (AUROC 0.991 (95%CI 0.975–1.0), *p* < 0.001). Abbreviations: REE_VCO2_ ^RQ 0.89^ = Resting Energy Expenditure (REE) based on volumetric carbon dioxide production (VCO_2_) measurements (REE_VCO2_) using assumed RQ of 0.89; REE_IC_ = REE measured by indirect calorimetry (IC).

**Figure 5 nutrients-14-04211-f005:**
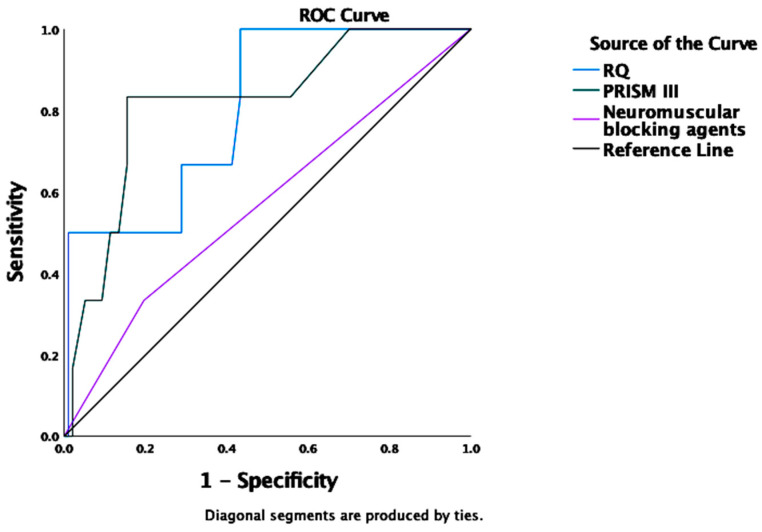
An Area Under the Receiver Operating Characteristic Curve (AUROC) for predicting REE_VCO2_ ^RQ 0.89^ overestimating REE_IC_ for more than +10%. Only the measured RQ (AUROC 0.804 (95%CI 0.643–0.966), *p* = 0.013) and a high PRISM III score (AUROC 0.819 (95%CI 0.646–0.992), *p* = 0.009) were strong predictors of REE_VCO2_ inaccuracy.

**Table 1 nutrients-14-04211-t001:** Demographic and clinical characteristics of the study population.

Characteristic	Variable	N = 107
Demographic	Age (years)	9.2 ± 5.3
	Sex (boy/girl)	75/32 (70.1%/29.9%)
	Body weight (kg)	35.9 ± 26
	Height (cm)	129 ± 29
	BMI (kg/m^2^)	18.7 ± 6
	z-score weight for age	0.33 (−1.5; 1.6)
	z-score height for age	−0.02 (−0.48; 0.66)
	z-score BMI for age	0.22 (−1.26; 1.68)
	Underweight	23 (21.5%)
	Normal BMI	47 (43.9%)
	Overweight	10 (9.3%)
	Obese	27 (25.2%)
Reasons for PICU admission	Respiratory failure	25 (23.4%)
	Sepsis	20 (18.7%)
	Surgical	9 (8.4%)
	Organ failure	2 (1.9%)
	Trauma	28 (26.2%)
	Neurologic	23 (21.5%)
Clinical data	PRISM score	11 (8; 15)
	TISS score	43 (36; 47)
	PELOD score	7 (3; 19)
	FiO_2_ (%)	35 (30; 50)
	pH	7.38 (7.34; 7.42)
	pO_2_ (mmHg)	112 (94; 121)
	pCO_2_ (mmHg)	35 (33.9; 39.3)
	HCO_3_ (mEq/L)	22.2 (19.0; 23.9)
	Heart Rate (bpm)	98 (78; 117)
	Respiratory rate (bpm)	20 (16; 28)
	Systolic Blood Pressure (mmHg)	94 (75; 110)
	Body Temperature (° Celsius)	37.4 (36.7; 38.1)
	Lactate (mg/dL)	14.1 (6.9; 31)
	Glucose (mg/dL)	105 (94; 121)
	Albumin (mg/dL)	3.2 (2.6; 3.6)
	C-Reactive Protein (mg/dL)	9.7 (2.2; 18)
	Vasoactive agents (yes) (%)	58 (54.24%)
	Sedatives and/or opioids > 2 (%)	91 (85%)
	Neuromuscular blocking agents (yes) (%)	23 (21.5%)
	Length of Stay (days)	14 (7; 24)
	Mechanical Ventilation (days)	12 (7; 18)
	Hospital Mortality	4 (3.7%)
Nutrition	Energy intake (kcal/day)	720 (480; 1000)
	Energy intake/IBW (kcal/kg/day)	24 (13.2; 42.8)
	Adequate feeding	38 (35.5%)
	Underfeeding	50 (46.7%)
	Overfeeding	19 (17.8%)

Continuous variables are reported as mean ± SD or median (interquartile range) as appropriate. Discrete variables are reported as the number and proportion (within brackets) of subjects with the characteristic of interest. Abbreviations: BMI = Body Mass Index; PRISM = Pediatric Risk of Mortality; TISS = Therapeutic Intervention Scoring System; PELOD = Pediatric Logistic Organ Dysfunction; IBW = Ideal Body Weight.

**Table 2 nutrients-14-04211-t002:** Comparison analysis between the VCO_2_ derived values and the resting energy expenditure measured by indirect calorimetry.

Variables	N = 107
VO_2_ (mL/min)	144.8 (105; 207.5)
VCO_2_ (mL/min)	115 (84.2; 175.4)
Respiratory Quotient	0.81 (0.75; 0.91)
REE_IC_ (kcal/day)	999 (703; 1416)
REE_VCO2_ (kcal/day)	910.8 (667; 1389)
REE_IC_/IBW (kcal/kg/day)	32.8 (24; 48.6)
REE_VCO2_/IBW (kcal/kg/day)	29.3 (29.3; 44)
Mean Bias ± SD (kcal/day) *	−72.73 ± 127
Limits of Agreement (kcal/day) *	−321.7 to 176.3
95% CI Lower-Upper (kcal/day) *	−92.8 to −49.9
Coefficient of Variation (%) *	174.7
Median of Differences (95%CI) (kcal/day) ^#^	−71.01 (−92.9; −49.9)
*p* value ^#^	<0.001
Cronbach’s alpha (kcal/day) ^^^	0.979 (0.970; 0.986)
*p* value ^^^	<0.001
REE_VCO2_ ± 10% of REE_IC_ **	52 (48.6%)
REE_VCO2_ > 10% of REE_IC_ **	6 (5.6%)
REE_VCO2_ < 10% of REE_IC_ **	49 (45.8%)
Normometabolic ^+^	20 (18.7%)
Hypometabolic ^+^	63 (58.9%)
Hypermetabolic ^+^	24 (22.4%)

Continuous variables are reported as mean ± SD or median (interquartile range) as appropriate. Discrete variables are reported as the number and proportion (within brackets) of subjects with the characteristic of interest. Abbreviations: VO_2_ = volumetric oxygen consumption VCO_2_ = volumetric carbon dioxide production; REE = Resting Energy Expenditure; IC = Indirect Calorimetry; REE_VCO2_ = REE based on VCO_2_ measurements alone; REE_IC_ = REE measured by IC; IBW = Ideal Body Weight; SD = Standard Deviation; CI = Confidence Interval.* Bland–Altman; ^#^ Wilcoxon matched pairs signed rank test; ^^^ Reliability by Cronbach’s alpha using the two-way mixed consistency, identical to the intraclass correlation coefficient (ICC); ^+^ Hypometabolic, hypermetabolic, and normometabolic are defined as REE_VCO2_ of <90%, >110%, and between 90% and 110% of basal metabolic rate as predicted by Schofield’s equation, respectively [20]. ** Clinically significant percentage error (REE_VCO2_–REE_IC_)/REE_IC_ (%). Statistical significance was considered for *p* < 0.05.

**Table 3 nutrients-14-04211-t003:** Comparison analysis between the VCO_2_ derived values and the resting energy expenditure measured by indirect calorimetry.

REE Estimation		REE (kcal/day)		Agreement—Precision *			Paired Differences—Variability ^#^				Accuracy ^		
Calculated REE (Reference)	Equation	Median	IQR (25th; 75th)	Mean Bias	SD	Limits of Agreement	Median of Differences	95% CI of Differences Lower-Upper	CV (%)	*p* Value	REE_VCO2_ < 10% of REE_IC_	REE_VCO2_ ± 10% of REE_IC_	REE_VCO2_ > 10% of REE_IC_
						N = 107							
REE_IC_ [3]	[3.941 × VO_2_ + 1.106 × VCO_2_] × 1440	999.0	(703; 1416)										
REE_VCO2_ [10]	5.5 × VCO_2_ (L/min) × 1440 [10]	910.8	(667; 1389)	−72.73	127.0	−321.7; 176.3	−71.01	−92.8; −49.9	174.7	<0.001	49 (45.8)	52 (48.6)	6 (5.6)
REE_VCO2_ [18]	5.534 × VCO_2_ (L/min) × 1440 [10]	916.4	(671; 1398)	−66.56	126.5	−314.4; 181.3	−64.02	−87.2; −41.8	190	<0.001	47 (43.9)	53 (49.5)	7 (6.5)
REE_VCO2 fixed RQ 0.89_ [19]	((5.5 × (VCO_2_/0.89)) + (1.76 × VCO_2_) − 26)	887.1	(642.5; 1367)	−96.24	126.8	−344.8; 152.3	−94.38	−116.6; −73.17	131.8	<0.001	58 (54.2)	46 (43)	3 (2.8)
REE_VCO2 fixed RQ 0.85_ [19]	((5.5 × (VCO_2_/0.85)) + (1.76 × VCO_2_) − 26)	920.5	(667; 1418)	−59.62	124.3	−303.3; 184	−64.73	−84.62; −41.49	208.5	<0.001	46 (43)	51 (47.7)	10 (9.3)
REE_VCO2 fixed RQ 0.80_ [19]	((5.5 × (VCO_2_/0.80)) + (1.76 × VCO_2_) − 26)	967	(701.1; 1489)	−8.70	124.0	−251; 234.4	−29.66	−46.21; 6.47	1427	0.332	21 (19.6)	61 (57)	25 (23.4)
REE_VCO2 measured RQ by IC_	((5.5 × (VCO2/RQ_IC_)) + (1.76 × VCO2) − 26)	973.5	(686.6; 1442)	−38.48	102.4	−239.1; 162.2	−24.15	−26.61; −13.48	266	<0.001	9 (8.4)	96 (89.7)	2 (1.9)

Continuous variables are reported as mean ± SD or median (interquartile range) as appropriate. Discrete variables are reported as the number and proportion (within brackets) of subjects with the characteristic of interest. Abbreviations: VO_2_ = volumetric oxygen consumption VCO_2_ = volumetric carbon dioxide production; REE = Resting Energy Expenditure; IC = Indirect Calorimetry; REE_VCO2_ = REE based on VCO_2_ measurements alone; REE_IC_ = REE measured by IC; RQ_IC_ = Respiratory Quotient measured by IC; IQR = interquartile range; SD = Standard Deviation; CI = Confidence Interval; CV = Coefficient of Variation.* Bland–Altman; ^#^ Wilcoxon matched pairs signed rank test; ^ Clinically significant percentage error (REE_VCO2_–REE_IC_)/REE_IC_ (%). Statistical significance was considered for *p* < 0.05.

**Table 4 nutrients-14-04211-t004:** Area Under the Curve for variables predicting REE_VCO2_ ^RQ 0.89^ inaccuracy by underestimating REE_IC_ for more than −10%.

Asymptotic 95% Confidence Interval					
Test Result Variable(s)	Area	Std. Error *	Asymptotic Sig. **	Lower Bound	Upper Bound
RQ	0.991	0.008	0.000	0.975	1.00
Age (years)	0.365	0.079	0.101	0.209	0.52
Sex	0.591	0.081	0.268	0.432	0.751
BMI z-score	0.565	0.086	0.432	0.396	0.734
Metabolic status	0.537	0.084	0.655	0.372	0.701
PRISM III	0.381	0.081	0.151	0.222	0.541
Heart rate	0.458	0.086	0.607	0.289	0.626
Number of sedatives-opiods	0.5	0.083	0.996	0.338	0.662
Vasoactive agents (yes)	0.538	0.082	0.641	0.377	0.700
Respiratory rate	0.542	0.084	0.607	0.378	0.707
Neuromuscular blocking agents (yes)	0.413	0.082	0.294	0.253	0.574

Abbreviations: RQ = Respiratory Quotient (measured by IC); BMI = Body Mass Index; PRISM = Pediatric Risk of Mortality; * Under the nonparametric assumption; ** Null hypothesis: true area = 0.5.

**Table 5 nutrients-14-04211-t005:** Area Under the Curve for variables predicting REE_VCO2_ ^RQ 0.89^ inaccuracy by overestimating REE_IC_ for more than +10%.

Asymptotic 95% Confidence Interval					
Test Result Variable(s)	Area	Std. Error *	Asymptotic Sig. **	Lower Bound	Upper Bound
RQ	0.804	0.082	0.013	0.643	0.966
PRISM III	0.819	0.088	0.009	0.646	0.992
Neuromuscular blocking agents (yes)	0.569	0.128	0.573	0.319	0.819

Abbreviations: RQ = Respiratory Quotient (measured by IC); PRISM = Pediatric Risk of Mortality; * Under the nonparametric assumption; ** Null hypothesis: true area = 0.5.

## Data Availability

The datasets generated and analyzed during the current study are not publicly available due the database is very extensive and includes data from other studies complementary to this but are available from the corresponding authors upon reasonable request.

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
