# Peer review of "External Validation with Accuracy Confounders of VCO2-Derived Predicted Energy Expenditure Compared to Resting Energy Expenditure Measured by Indirect Calorimetry in Mechanically Ventilated Children"

_nutrients, 2022, doi:10.3390/nu14194211_

Round 1

Reviewer 1 Report

In the present MS External and internal validation of VCO2-derived predicted energy expenditure compared to resting energy expenditure measured by indirect calorimetry in mechanically ventilated children”

Briassoulis and coworkers presented the possibility of using REEVCO to predict REEIC.

Overall, I think that the manuscript is straight forward, and easy to read. However, there are some weak points

1.     Sepsis and organ failure also induce metabolic disorder. Will be results changed if exclude data form sepsis and organ failure?

2.  Since energy intake was not associated with REEVCO2 inaccuracy, is it possible that RQ various with diseases?

3.  Authors measuring REEvco2 during the first 24 hours of admission. Please provide the reason and will RQ change over-time?

Author Response

In the present MS  External and internal validation of VCO2-derived predicted energy expenditure compared to resting energy expenditure measured by indirect calorimetry in mechanically ventilated children”

Briassoulis and coworkers presented the possibility of using REEVCO to predict REEIC.

Overall, I think that the manuscript is straight forward, and easy to read. However, there are some weak points

1.     Sepsis and organ failure also induce metabolic disorder. Will be results changed if exclude data form sepsis and organ failure?

Point 1: Thank you for pointing out this misconception. We have not excluded data form sepsis and organ failure. In table 1 among clinical diagnoses are Respiratory failure, Sepsis, Surgical, Organ failure. We have now clarified in methods that metabolic disorders are not temporary metabolic derangements  secondary to critical illnesses such as organ failure but: (2) inborn errors of metabolism or primary endocrine disorders;

2.  Since energy intake was not associated with REEVCO2 inaccuracy, is it possible that RQ various with diseases?

Point 2: Thank you for this question. We did not find any relation of RQ with illness severity or diagnosis but only with lactate levels. In the results we showed that "Bivariate analysis showed that REEVCO2 -REEIC difference (rs=0.35, p=0.013) and RQ (rs=0.32, p=0.022) correlated with lactate, but not with BMI z-scores, age, metabolic status, TISS, PRISM, PELOD, heart or respiratory rate, blood pressure, temperature, blood gases, glucose, albumin, C-reacting protein, or energy intake". In discussion we verify that "Except for lactate, we did not find any correlation between RQ and clinical, metabolic, or nutrition indices." Also, presenting findings of other studies, we discuss that "In accordance with our previous research, this study showed that RQ is highly variable and unpredictable in mechanically ventilated children [29], limiting its validity as an indicator of energy substrate oxidation [30], misleading energy provision targets [31], and enhancing the risk of underfeeding and overfeeding [32]." Similarly, "They confirmed that RQ is neither a reliable indicator of the feeding status nor strongly associated with non-nutritional factors and concluded that REEVCO2 could not be considered an alternative to REEIC."

3.  Authors measuring REEvco2 during the first 24 hours of admission. Please provide the reason and will RQ change over-time?

Point 3: We would like to follow up with longitudinal studies, but this was technically not possible at that time because of limited sources. We organize now a second-phase longitudinal study. In limitations of the study, we acknowledged that "Also, this is a prospective cross-sectional study, while the timing of the indirect calorimetry measurements reflects the acute only metabolic phase of illness. Although insignificant differences have been previously demonstrated during the critical first week of critical illness [12], other studies showed a pattern of early longitudinal repression of bioenergetics, the persistence of which is associated with poor outcomes [7,30,33].

Reviewer 2 Report

The authors present an analysis of the agreement of VCO2 predicted resting energy expenditure to energy expenditure calculated from indirect calorimetry on a sample of n = 107 mechanically ventilated children admitted to the ICU. They find that REEVCO2 has high variability and bias compared to IC measured REE which cannot be attributed to individual respiratory quotient or any metabolic, anthropometric or other risk factors. The authors conclude that REE VCO2 is not an adequate replacement for REE IC.

The paper is well written and the analyses well conducted. The underlying research hypothesis is interesting and clinically relevant. However I have some comments about the data presentation, see below.

·        I feel that the overall message of the manuscript could be a tad more constructive. So the REEVCO2 is not a good proxy – what would potentially be one? Since the IC measures are not available for all patients due to money/logistics, shouldn’t efforts be directed to find more accessible measures that work well? I guess some kind of outlook and suggestions would help here.

·        The main predictor of REEVCO2 accuracy is individual RQ. But using individual instead of fixed RQ did not improve overall performance. And overall performance was not associated with any demographic, metabolic or other risk factor. So what is the reason that REEVCO2 doesn’t work so well? Could the underlying equation maybe be improved by a calibration factor or similar, is there any research/references on that?

·        The term “internal validation” seems misleading from a statistical point of view, since this is rather an analysis of potential predictor variables than an internal validation of a prediction model. I suggest to rephrase.

·        Abstract: Define abbreviation RQ

·        Please elaborate the flow of  N =486 to N =121 according to exclusion criteria.

·        Exclusion: metabolic or endocrine disorders. How many patients were excluded due to this criterion? I could imagine that especially for these patients, precise measurements would be needed, so it might be worthwhile to give that as an outlook

·        Ref 4 and 5 are identical, please correct.

·        Figure 3 and 4 are hard to decipher. Please provide an overview table containing the AUC values and Cis for each curve. Color scheme should be the same between the two figures.

·        Table 1, Mortality: does this denote mortality during the hospital stay, or were patients followed up?

·        Table 2, how was normo/hypo/hypermetabolism defined?

·        L332f: please provide arguments and references that justify calling the work of Rousing et al (please correct the spelling) “unreliable”. Or rephrase.

·        L341: Who is the “control group”? please explain/rephrase

·        L378f “that this is a factor that did not need to be considered” – what does “this” refer to in this context?

Author Response

The paper is well written, and the analyses well conducted. The underlying research hypothesis is interesting and clinically relevant. However, I have some comments about the data presentation, see below.

Point 1:

I feel that the overall message of the manuscript could be a tad more constructive. So the REEVCO2 is not a good proxy – what would potentially be one? Since the IC measures are not available for all patients due to money/logistics, shouldn’t efforts be directed to find more accessible measures that work well? I guess some kind of outlook and suggestions would help here.

Response 1:. Thank you for pointing out this critical question. We have approached the problem throughout the study, examining possible outlets to the deadlock of alternatives to IC. There is no one. In our conclusions, we state that "Since the accuracy of REEVCO2 may not be improved using various arbitrary or individual RQ values, online VO2 measurements are critical in accurately calculating REE in mechanically ventilated patients. But VO2 is currently available only when using calorimeters. "Similarly, promising predictive equations have repeatedly been shown to be grossly inaccurate. In our opinion, the only way forward is to make calorimeters widely available and cost-effective. We agree with the reviewer that some kind of outlook and suggestions would help and added the following sentence at the end of the conclusions " A new generation of user-friendly calorimeters, cost-effective, incorporated into venti-lators' hardware and software look like a one-way street to overcome current limitations in reliably measuring real-time REE in an intensive care setting.    ."

Point 2:

The main predictor of REEVCO2 accuracy is individual RQ. But using individual instead of fixed RQ did not improve overall performance. And overall performance was not associated with any demographic, metabolic or other risk factor. So what is the reason that REEVCO2 doesn’t work so well? Could the underlying equation maybe be improved by a calibration factor or similar, is there any research/references on that?

Response 2:.Thank you for discussing this problem. To assess this problem, we used different RQs, which had been used in studies on adults and children, and also individual RQs. We have examined and discussed extensively all available hypotheses and other studies' results. Finally, we were able to show for the first time that the REEVCO2 equation cannot be improved because the equation lacks the individuals' VO2 values. Without VO2 there is no remedy to the equation. But VO2 is only available in the modified Weir's equation used by IC. In line 482 we discuss. "Because the time constant for VO2 equilibration is much shorter (2-3 min) than the VCO2 (10-20 min), variations in ventilation will affect differently the REEIC, incorporating the VO2IC in the equation, compared to the REEVCO2, missing the breath-by-breath metabolic marker VO2IC quick measurements [24]. The wide scattering of VO2IC values in our control group verifies the vulnerability of the REEVCO2 in estimating the patient's metabolism changes, precluding its use as an alternative to REEIC in mechanically ventilated patients."

Point 3:

The term “internal validation” seems misleading from a statistical point of view, since this is rather an analysis of potential predictor variables than an internal validation of a prediction model. I suggest to rephrase.

Response 3: As suggested by the reviewer, we have rephrased the term “internal validation” as follows:

External validation with accuracy confounders of VCO2-derived predicted energy expenditure compared to resting energy expenditure measured by indirect calorimetry in mechanically ventilated children

Point 4:

Abstract: Define abbreviation RQ

Response 4: Defined.

Point 5:

Please elaborate the flow of N =486 to N =121 according to exclusion criteria.

Response 5: Thank you for this suggestion. We have now elaborated the flow chart of enrolled patients.

Point 6:

Exclusion: metabolic or endocrine disorders. How many patients were excluded due to this criterion? I could imagine that especially for these patients, precise measurements would be needed, so it might be worthwhile to give that as an outlook

Response 6: The number of excluded patients due to this criterion is now depicted in the flow chart. Patients with endogenous metabolic or endocrine disorders (inborn errors of metabolism, diabetic ketoacidosis) were excluded because of unpredicted or complicated metabolic derangements or tachypnea or non-ventilation status (DKA). Temporary metabolic changes secondary to critical illness, such as electrolyte or glucose variability, were not included in this exclusion criterion.

Point 7:

Ref 4 and 5 are identical, please correct.

Response 7: Thank you for having noticed this. Reference 5 has now been corrected.

Point 8:

Figure 3 and 4 are hard to decipher. Please provide an overview table containing the AUC values and Cis for each curve. Color scheme should be the same between the two figures.

Response 8: Thank you for the suggestion, we have provided tables 4 and 5 containing the AUC values and Cis for each curve. Color schemes have now been synchronized between Figures 3 and 4.

Point 9:

Table 1, Mortality: does this denote mortality during the hospital stay, or were patients followed up?

Response 9: Hospital mortality. Now clarified in the table.

Point 10:

Table 2, how was normo/hypo/hypermetabolism defined?

Response 10: In the methods section 2.6 we define that "Basal metabolism was calculated based on the Schofield equation [21]. Hypometabolism and hypermetabolism were defined as measured REE of <90% and >110% of basal metabolic rate as predicted by Schofield's equation, respectively [21]." This definition has now been briefly added in the Table 2 footnotes.

Point 11:

L332f: please provide arguments and references that justify calling the work of Rousing et al (please correct the spelling) “unreliable”. Or rephrase.

Response 11: Thank you for having noted this weakness and typo. We have now corrected the spelling and rephrased the misleading word. Large variations in minute ventilation increase the discrepancy between metabolic production and pulmonary uptake or CO2 excretion, increasing the VCO2 variability.

Point 12:

L341: Who is the “control group”? please explain/rephrase

Response 12: Control group is defined in methods, 2.5 "As a control, we calculated equation 3 using the IC-derived individual RQ (RQIC) along with the individual VCO2 values, resulting in a VO2 alone REEVCO2 - REEIC difference (Equation 3). We have now added the abbreviation definition IC-derived individual RQ (RQIC).

Point 13:

L378f “that this is a factor that did not need to be considered” – what does “this” refer to in this context?

Response 13: This is the feeding status mentioned in the previous sentence. This has now been replaced by the real factor: "the feeding status".

Round 2

Reviewer 2 Report

I thank the authors for providing a revised version of their work. My comments were sufficiently addressed.